# Mitochondrial Disruption by Amyloid Beta 42 Identified by Proteomics and Pathway Mapping

**DOI:** 10.3390/cells10092380

**Published:** 2021-09-10

**Authors:** Patricia Sinclair, Ancha Baranova, Nadine Kabbani

**Affiliations:** 1Interdisciplinary Program in Neuroscience, School of Systems Biology, George Mason University, Fairfax, VA 22030, USA; psincla2@gmu.edu; 2School of Systems Biology, George Mason University, Manassas VA 20110, USA; abaranov@gmu.edu

**Keywords:** Alzheimer’s, amyloid, bioenergetics, mitochondria, prohibitin, proteomics

## Abstract

Alzheimer’s disease (AD) is marked by chronic neurodegeneration associated with the occurrence of plaques containing amyloid β (Aβ) proteins in various parts of the human brain. An increase in several Aβ fragments is well documented in patients with AD and anti-amyloid targeting is an emerging area of therapy. Soluble Aβ can bind to various cell surface and intracellular molecules with the pathogenic Aβ_42_ fragment leading to neurotoxicity. Here we examined the effect of Aβ_42_ on network adaptations in the proteome of nerve growth factor (NGF) differentiated PC12 cells using liquid-chromatography electrospray ionization mass spectrometry (LC-ESI MS/MS) proteomics. Whole-cell peptide mass fingerprinting was coupled to bioinformatic gene set enrichment analysis (GSEA) in order to identify differentially represented proteins and related gene ontology (GO) pathways within Aβ_42_ treated cells. Our results underscore a role for Aβ_42_ in disrupting proteome responses for signaling, bioenergetics, and morphology in mitochondria. These findings highlight the specific components of the mitochondrial response during Aβ_42_ neurotoxicity and suggest several new biomarkers for detection and surveillance of amyloid disease.

## 1. Introduction

Alzheimer’s disease (AD) is the most common cause of human dementia and a major public health problem. Commonly prescribed drugs for treatment include acetylcholinesterase (AChE) inhibitors, which lessen cognitive symptoms without halting neurodegeneration [1]. A recently approved monoclonal antibody, aducanumab (Aduhelm), which specifically targets Aβ oligomers, appears to slow disease progression [2]. In clinical studies, however, aducanumab was also associated with transient swelling of the brain and hemorrhaging, thus increasing the urgency for a better understanding of the cellular effects of Aβ.

Amyloid plaques and neurofibrillary tangles (NFT) of aggregated hyperphosphorylated tau are the primary histopathological lesions of AD [3]. Amyloid plaques consist of highly structured Aβ peptides generated by β- and γ- secretase cleavage of APP in cells [4]. High levels of Aβ_42_ were shown to contribute to neural damage in early and late stages of AD [5]. Soluble, low molecular weight, Aβ_42_ oligomers contribute to cell damage and neurotoxicity [6], with the pathogenicity of amyloid proteins developing well before the emergence of measurable cognitive symptoms. In fact, memory deficits in mice correlate with the early stages of Aβ_42_ oligomer formation during Aβ deposition [7].

A mechanistic understanding of Aβ_42_ mediated neurotoxicity was obtained in cultured neural cells exposed to various concentrations of the amyloid protein [8]. Several subcellular and molecular targets of Aβ_42_ toxicity were identified [9,10]. In hippocampal neurons, Aβ_42_ peptides tend to aggregate within lipid rafts and drive Ca^2+^-mediated neurotoxicity (Abramov et al., 2004). Aβ_42_ oligomers also induce mitochondrial dysfunction and oxidative stress in neurons through calcium mediated neurotoxicity [11]. In addition, various cell surface receptors can bind to Aβ_42_, including metabotropic glutamate receptors (mGluR5), nicotinic acetylcholine receptor (α7nAChR), *N*-methyl-d-aspartic acid receptor (NMDAR), β-adrenergic receptor (β-AR), erythropoietin-producing hepatoma cell line receptor (EphR), and paired immunoglobulin-like receptor B (PirB) [12,13].

In this study, we examined whole cell proteomic adaptations to Aβ_42_ (Aβ_42_ proteome) in PC12 cells, which are an established model system for amyloid cytotoxicity [14]. Our findings underscore the role of mitochondrial responses to Aβ_42_ and delineate specific pathways for ion transport, calcium signaling, and energy production that may contribute to the early stages of AD.

## 2. Materials and Methods

### 2.1. Cell Culture and Treatment

Pheochromocytoma (PC12) adherent cells (ATCC^®^ CRL-1721.1™) were proliferated in RPMI-1640 (ATCC), which was supplemented with 10% horse serum (HS), 5% fetal bovine serum (FBS), and 1% pen/strep on collagen (50 μg/mL) at 37 °C and 5% CO_2_. Cells were differentiated using nerve growth factor (NGF, 200 ng/mL, Millipore, Burlington, MA, USA) in RPMI-1640, 2% HS, 1%FBS, and 0.2% pen/strep for 24 h before treating with 100 nM Aβ_42_. Soluble oligomeric Aβ_42_ peptides were prepared as described in Arora et al. [15]. These preparations were shown to yield a stable oligomeric form of Aβ_42_. Daily media and treatment changes ensured consistent exposure to Aβ_42_ [16]. After three days, cells were solubilized by the addition of a 0.1% Triton X-100 lysis buffer (Triton X-100, 1 M Tris HCl, 1.5 M NaCl, 0.25 M EDTA, and 10% glycerol, in the presence of protease inhibitors (Complete Mini, Roche, Basel, Switzerland) in 10 mL of lysis buffer) as described [17]. The protein concentration was determined by a Bradford assay.

### 2.2. Liquid-Chromatography Electrospray Ionization Mass Spectrometry

Sample preparation for proteomics was conducted as described in [18]. Briefly, 100 μg of protein was precipitated by the addition of acetone 100 μL at 4 °C for 5 min and then centrifuged for 3 min at 16,000× *g*. The pellet was sequentially treated with 8 M urea, 1 M dithiothreitol, 0.5 M iodoacetamide, and 2 μL (0.5 μg/μL) trypsin in 0.5 M ammonium bicarbonate, then incubated at 37 °C for 5 h to denature, reduce, alkylate, and digest proteins. The sample was desalted using C-18 ZipTips (Millipore), dehydrated in a SpeedVac for 18 min, and reconstituted with 0.1% formic acid for a final volume of 20 μL for liquid-chromatography electrospray ionization mass spectrometry (LC-ESI MS/MS). LC-ESI MS/MS was performed using an Exploris Orbitrap 480 equipped with an EASY-nLC 1200HPLC system (Thermo Fischer Scientific, Waltham, MA, USA). Peptides were separated using a reversed-phase PepMap RSLC 75 μm i.d by 15 cm long with a 2 μm particle size C18 LC column (Thermo Fisher Scientific, Waltham, MA, USA), eluted with 0.1% formic acid and 80% acetonitrile at a flow rate of 300 nL/min. After one full scan (60,000 resolution) from 300 *m*/*z* to 1200 *m*/*z*, high abundance peptides were selected in a data-dependent fashion for fragmentation by high-energy collision dissociation (HCD) with a normalized collision energy of 28%. Enabled filters included EASY-IC internal mass calibration, monoisotopic precursor selection, and dynamic exclusion (20 s). Peptide precursor ions with charge states from +2 to +4 were included. All samples were run in triplicate.

### 2.3. Protein Identification and Quantification

Proteome Discoverer *v*2.4 (Thermo Fisher Scientific, Waltham, MA, USA) was used to identify proteins using the SEQUEST HT search engine to compare MS spectra to the *Rattus norvegicus* Rat_NCBI2016 database under the following parameters: mass tolerance for precursor ions = 5 ppm; mass tolerance for fragment ions = 0.05 Da; and cut-off value for the false discovery rate (FDR) in reporting peptide spectrum matches (PSM) to the database = 1%. The abundance ratios and p-values were calculated by precursor ion quantification in Proteome Discoverer *v*2.4. Non-treated NGF differentiated PC12 cells were used as a control.

### 2.4. Gene Set Enrichment Analysis (GSEA) and Genomic Enrichment Analysis (GEA) 

Bioinformatic analysis was performed on filtered dataset proteins obtained from Proteome Discoverer *v*2.4 with a quantifiable spectra signal profile in at least two out of the three replicates. Proteins differentially impacted by the treatment condition were identified based on the calculated protein abundance ratio (fold-change) *p* < 0.05 using Proteome Discoverer. Proteins were identified by Entrez ID gene identifiers in NCBI. 

For GSEA or GEA, protein identifiers, abundance ratios, and individual *p*-values were uploaded into Pathway Studio (PS) (www.pathwaystudio.com, accessed 2 March 2021 through 20 June 2021). GSEA was performed against the Gene Ontology (GO) pathways stored in the PS environment.

### 2.5. Statistics

One-way analysis of variance (ANOVA), followed by Benjamini-Hochberg post-hoc, was used to calculate *p*-values within Proteome Discoverer *v*2.4. Statistical significance of GO pathways and GEA, analyzed in Pathway Studio, was obtained using a Mann-Whitney U test. *p*-values of < 0.05 were considered significant. The data was organized and presented using the R package, ggplot2 [19], and Excel. 

## 3. Results

### 3.1. Whole-Cell Proteome Analysis of PC12 Cells Exposesd to Aβ_42_

To uncover molecular mechanisms of Aβ_42_ mediated neurotoxicity, NGF differentiated PC12 cells were treated with 100 nM Aβ_42_ prepared and presented in a manner consistent with the formation of pathogenic oligomeric amyloid peptides [20]. PC12 cells were exposed to three days of Aβ_42_, then analyzed using LC-ESI-MS/MS. In these experiments, NGF differentiated cells not exposed to Aβ_42_ were used as the control condition. Using MS proteomic spectra identification and label-free precursor ion quantification for relative abundance [21] in Proteome Discoverer *v*2.4, we identified proteins significantly impacted by the presence of Aβ_42_. Pathway Studio (PS) and Gene Ontology (GO) informatic analysis was used to deduce significant molecular pathways and cellular components of the Aβ_42_ response (Figure 1).

MS proteomic analysis identified a total of 4206 known proteins within our cells; a complete list has been deposited in FigShare dataset (10.6084/m9.figshare.15157257). Of those, a subset of 316 proteins was identified as significantly altered by Aβ_42_ relative to controls (*p* < 0.05) and are assigned the label “Aβ_42_ proteome” in the study. As shown in Figure 2A (Appendix A), within the Aβ_42_ proteome 226 proteins (72%) were decreased, while 90 proteins (28%) were increased. GO analysis of the Aβ_42_ proteome revealed an impact on mitochondrial processes (Figure 2B–D) consistent with earlier findings in neurons [22].

### 3.2. The Aβ42 Proteome Revolves around Mitochondria Function

Using GO analysis, 199 proteins (63%) within the Aβ_42_ proteome were classified as membrane-associated molecules (Figure 3A, Appendix A). Further subcellular analysis localized these membrane proteins to mitochondria (28%), nucleus (10%), endoplasmic reticulum (10%), Golgi apparatus (1%), endosomes (3%), cell surface (2%), and/or more than one compartment (multi-compartment) (23%) (Figure 3B). Interestingly, the majority of multi-compartment proteins (76%) within the Aβ_42_ proteome, showed mitochondrial and nuclear placement (Figure 3C). For example, prohibitin 2 (PHB2), which was significantly downregulated by Aβ_42_, is known to shuttle between the mitochondria and nucleus and regulate transcription [23].

### 3.3. Gene Set Enrichment Analysis of Amyloid-Associated Pathways 

Using Gene Set Enrichment Analysis (GSEA) in Pathway Studio (https://www.pathwaystudio.com, accessed in 2 March 2021) [24], Aβ_42_ proteome functions were extracted and analyzed. A bioinformatic assessment of all significantly altered proteins within the Aβ_42_ proteome (Appendix A), identified 30 specific enriched GO pathways for the Aβ_42_ proteome (Figure 4 and Table 1). A trend for engaging adenine nucleotide transmembrane transport (ADP/ATP) and mitochondrial ribosomal proteins (mitoribosome) in the presence of amyloids emerged. This bioinformatic analysis complements previous findings, which showed an impact of amyloid on mitochondrial protein expression [25] and suggests a specific role for proteins, such as SLC25A5, in oxidative phosphorylation (OXPHOS) amyloid toxicity.

Earlier studies demonstrated an ability of Aβ [26] and α-synuclein [27], to impact mitochondrial trafficking and function. Consistently, we detected significant changes in the expression of many mitochondrial proteins (95 down-regulated; and 13 up-regulated) within the Aβ_42_ proteome dataset (Figure 5A, Appendix A). GO analysis showed that these proteins were important for mitochondrial membrane organization, inner structural scaffolding, and mitoribosome function (Figure 5B). In particular, a significantly downregulated set of solute carriers (SLC) proteins, including SLC25A4 and SLC25A5, are known to mediate nucleotide (ADP/ATP) transport across the mitochondrial membrane [28]. These proteins are also known to regulate nuclear function (Figure 5C, Appendix A) [29].

### 3.4. An Analysis of Aβ_42_ Proteome Associated Genes and Possible Relevance to Human Disease 

Genomic enrichment analysis (GEA) of the human genome was performed on the Aβ_42_ proteome dataset using PS. Genes that encode components of the Aβ_42_ proteome were found randomly distributed throughout the human genome. Interestingly, ~70% of the Aβ_42_ proteome appeared to localize to gene regions of disease, including rare copy number variation (e.g., microdeletions and/or duplications). A complete list of the GEA results is presented in Appendix A and lists all genes, chromosomal location, and human disease phenotype associated with the Aβ_42_ proteome dataset. Figure 6, summarizes GEA of genes associated with mitochondrial proteins within the Aβ_42_ proteome. As shown, our results suggest a link between amyloid-associated cellular responses and various genetic factors that contribute to human disease. For example, chromosome 1p36 deletion syndrome appears impacted by amyloid-sensitive proteins encoded by the *SDHB* and *UQCRHL* gene, which both reside within that gene region and can critically contribute to cell energetics. In fact, 1p36 deletion syndrome was shown to associate with congenital cardiac abnormalities and/or developmental and intellectual deficits [30]. Another genomic region of interest resides on chromosome 17q21, housing both the *BRCA1* and *PHB* genes, whose products are estrogen receptor (ER) sensitive [31,32]. In this case, *PHB* encodes the mitochondrial protein prohibitin that is significantly decreased within our dataset, suggesting a link between estrogen and amyloid responses in cells.

### 3.5. Proteomic Mechanisms of Aβ_42_ Cellular Calcium Signaling

Amyloid proteins were shown to increase intracellular calcium levels, leading to calcium-mediated neurotoxicity [22,33]. Aβ_42_ mediates cellular calcium entry through binding to cell surface receptors, such as the α7 nACh [12,13]. Therefore, the subcellular distribution of calcium binding molecules (i.e., proteins with a known calcium binding domain) and more broadly, calcium associated molecules (i.e., those with known regulation by calcium) was analyzed within the Aβ_42_ proteome. As shown in Figure 7, Aβ_42_ treatment was associated with a noticeable impact on cellular calcium signaling through the regulation of both calcium binding and associated proteins. GO domain mapping indicated that impacted proteins localize to various subcellular compartments and can traffic between important organelles. For example, FKBP1A, which was significantly up-regulated (fold-change = 4.632, *p* = 2.37 × 10^−16^) in the presence of Aβ_42_, is a calcium binding *cis*-trans isomerase that modulates ryanodine receptor (RyR) activity and protein folding in the ER. Here, FKBP1A appears to participate in RyR mediated calcium flux from the ER to the mitochondria at the mitochondrial associated membrane (MAM) [34,35] (Figure 7 and Appendix A). Overall, our proteomic findings support existing evidence on the direct ability of amyloid peptides to disrupt intracellular calcium activity in neural cells [36], and suggest a multi-compartment model for the impact of amyloids on calcium homeostasis.

## 4. Discussion

Neurodegeneration in the course of AD is suspected to stem from alterations in amyloid precursor protein (APP) processing, resulting in the elevation of a pathogenic Aβ_42_ peptide in the brain and cerebrospinal fluid (CSF) [37]. We investigated the direct impact of this pathogenic amyloid peptide on cells using whole cell proteome analysis and GO analysis. We found that Aβ_42_ exposure preferentially impacted proteins associated with the mitochondria including its known interactions with other cellular organelles. We also identified other pathways of known importance for amyloid protein processing including two subunits of the biogenesis of lysosome-related organelle complex 1 (BLOC1S2 and BLOC1S6) that are involved in endosome recycling, another mechanism implicated in AD [38,39,40]. These proteins are consistent with the concurrent activity of lysosome and autophagy pathways that known to be activated during by amyloid protein degradation in neural cells [38]. 

The Aβ_42_ proteome reveals a mechanistic framework for understanding how amyloid peptides disrupt cell bioenergetics, calcium homeostasis, and the shape and localization of mitochondrion [41]. As depicted in Figure 8, our data proposes an important impact on mitochondrial cristae scaffolding proteins and disruption to energy management through proteins such as prohibitin. Specifically, the energy producing cellular machinery, including components of metabolite transport, tricarboxylic acid cycle (TCA), the electron transport chain (ETC) (complexes I–IV), and the ATP synthase, all appear to be impacted by Aβ_42_. The ETC, responsible for creating and maintaining the vital H^+^ gradient across the mitochondrial membrane, upon exposure to Aβ_42_ is likely perturbed at the H^+^ generating complex (I, III and IV) as based on proteomic evidence of changes to components of complex II during Aβ_42_ exposure. Here, a down-regulation in succinate dehydrogenase (SDBH) is also predicted to disrupt the TCA cycle and its link to the ETC [42]. This proposed failure of the ETC should yield a drop in mitochondrial ATP synthesis and drive ROS production, leading to disruption in membrane potential (Δψ_m_) [43]. This model is consistent with the observed features of amyloid toxicity [41].

The Aβ_42_ proteome is also deficient in a number of other mitochondrial proteins essential for cell survival and synaptic function. For example, Aβ_42_ was found to decrease the expression of the pyruvate carrier MPC2, a component of the pyruvate dehydrogenase complex that transports pyruvate into the mitochondrial matrix [44]. In addition, Aβ_42_ was found to negatively impact the levels of proteins vital for the exchange of ATP/ADP, phosphate, and various metabolites across the mitochondrial membrane. Observed changes in several key SLC25 family transport proteins, including SLC25A4 and SLC25A5, necessary for the exchange of ATP/ADP [45], and SLC25A3, involved in phosphate transport during ATP production, suggest several mitochondrial biomarker proteins in early AD detection.

## Figures and Tables

**Figure 1 cells-10-02380-f001:**
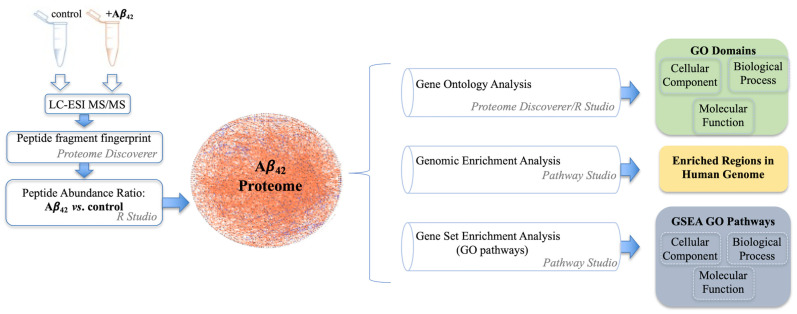
A flowchart summary of the mass spectrometry and bioinformatic analysis approach.

**Figure 2 cells-10-02380-f002:**
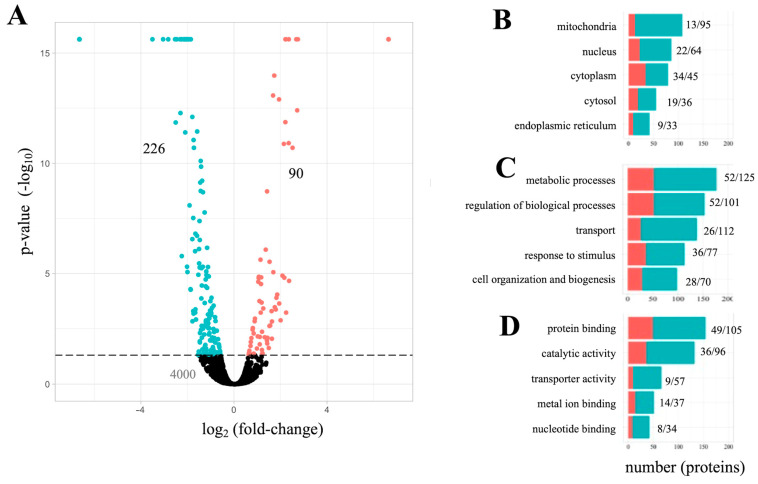
The Aβ_42_ proteome and GO impacted process. (**A**) each point on the plot represents 1 of 4206 identified proteins. The dotted line indicates the threshold of statistical significance (−log10 *p*-value). Green: 226 decreased proteins; red: 90 increased proteins. (**B**–**D**) the total number of significantly impacted proteins per GO domain: cellular component (**B**); biological process (**C**); and molecular function (**D**).

**Figure 3 cells-10-02380-f003:**
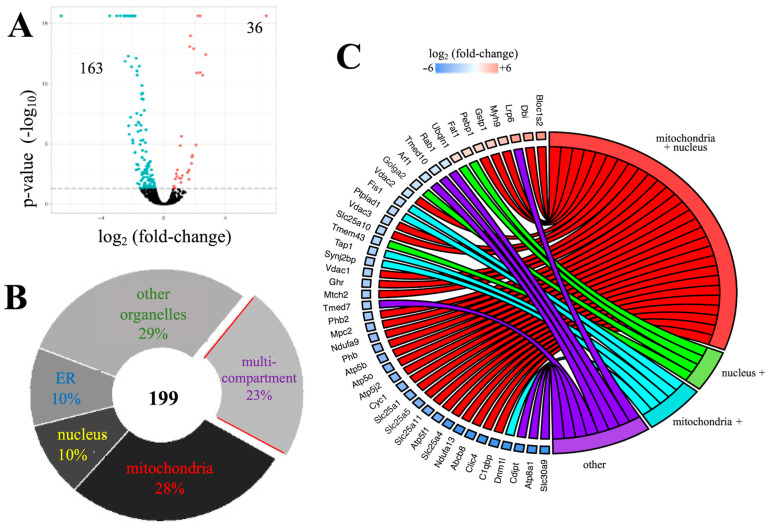
Subcellular distribution of membrane-associated components of the Aβ_42_ proteome. (**A**) the distribution of all 199 membrane associated proteins. (**B**) GO based subcellular allocation of these proteins shows: 28% mitochondria; 10% nucleus; 10% ER; 29% other organelle (e.g., proteasomes); 23% multi-compartment. (**C**) multi-compartment proteins show high mitochondrial and/or nuclear association.

**Figure 4 cells-10-02380-f004:**
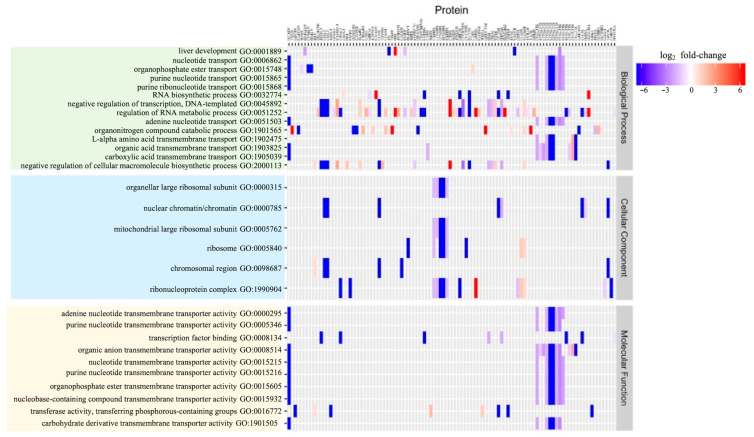
GSEA of the Aβ_42_ proteome highlights the contribution of mitochondrial pathways. A heatmap of GSEA within primary GO domains shows 30 enriched pathways within the Aβ_42_ proteome. The extent of protein abundance shift within Aβ_42_ treated cells (*vs*. control) is indicated.

**Figure 5 cells-10-02380-f005:**
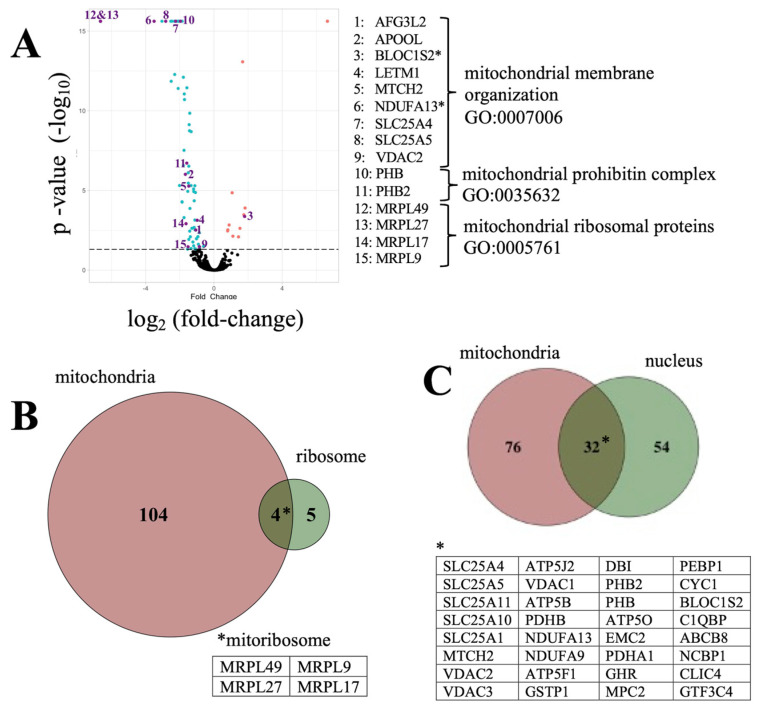
Mitochondrial subcomponents of the Aβ_42_ proteome. (**A**) the distribution of significantly altered proteins within Aβ_42_ proteome showing impacted subcomponents of mitochondrial organization and function according to GO analysis. The dotted line indicates the threshold to statistical significance (−log_10_ *p*-value). Green: 95 decreased proteins; red: 13 increased proteins. (**B**) the extent of mitochondria localized ribosomal proteins (mitoribosome) within Aβ_42_ proteome. (**C**) overlap between nuclear and mitochondrial proteins in the Aβ_42_ proteome. * mitoribosome proteins.

**Figure 6 cells-10-02380-f006:**
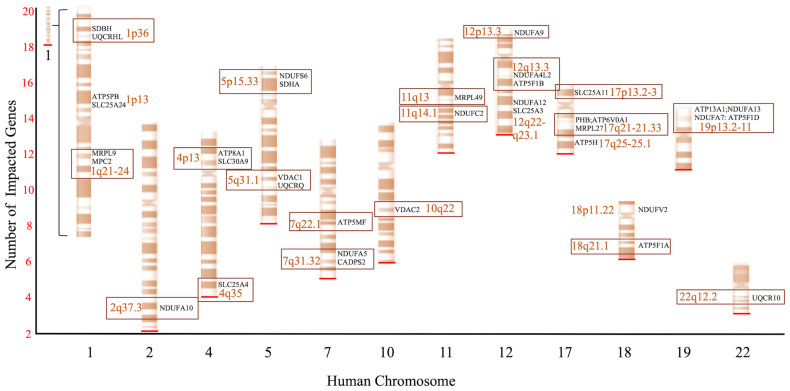
Genomic analysis of mitochondrial protein impacted by Aβ_42_. Genome enrichment analysis (GEA) was used to examine the number and location of genes that encode mitochondrial components of the Aβ_42_ proteome. The red line indicates the total number of impacted genes/chromosome. Boxed are chromosomal regions of copy number variation that correlate with human disease. Chromosome 1, which contains 18 impacted genes, has been adjusted for size.

**Figure 7 cells-10-02380-f007:**
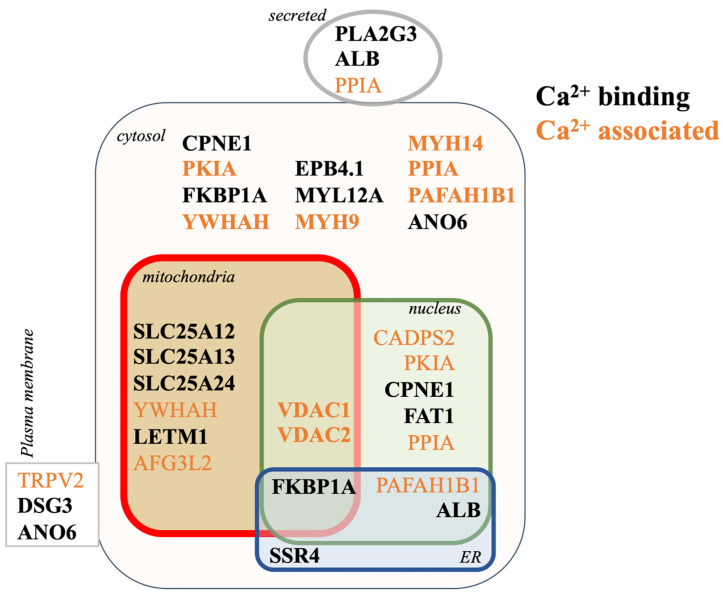
Cellular mechanisms of Aβ_42_ mediated calcium signaling. The subcellular distribution of calcium binding (black) and calcium associated (orange) proteins impacted by Aβ_42_.

**Figure 8 cells-10-02380-f008:**
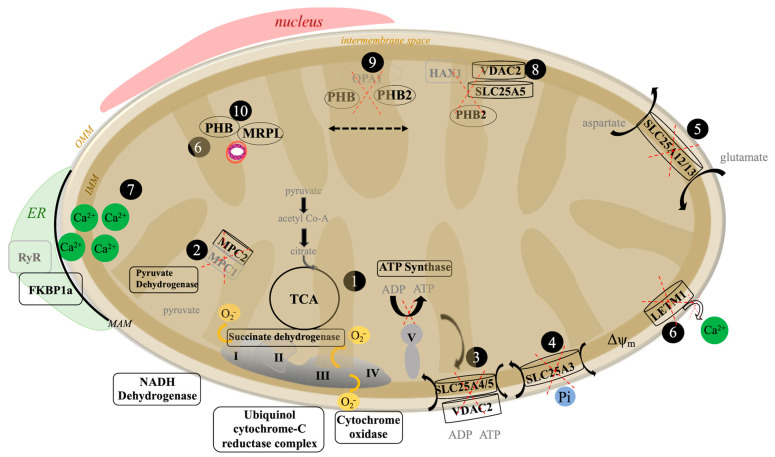
A model of mitochondrial disruption by Aβ_42_ based on our proteomic findings of specifically altered mitochondrial proteins (**bold**). 1: bioenergetics (TCA and ETC); 2: pyruvate dehydrogenase complex; 3: ATP/ADP exchange; 4: inorganic phosphate carrier; 5: glutamate/aspartate carrier; 6: LETM1 calcium transport; 7: FKBP1a stems calcium influx from ER; 8: anti-apoptotic HAX1 factor localized to the inter-membrane space through interaction with PHB2. 9: OPA-prohibitin complex stabilizes cristae; and 10: nucleoid stabilized to the IMM by PHB.

**Table 1 cells-10-02380-t001:** GSEA identified GO pathways enriched with the Aβ_42_ proteome data set.

GO ID #	Pathway Name	*p*-Value	# Entities Overlap
BIOLOGICAL PROCESS			
GO:0051503	adenine nucleotide transport	2.41 × 10^−2^	8
GO:0015868	purine ribonucleotide transport	2.41 × 10^−2^	8
GO:0015865	purine nucleotide transport	2.41 × 10^−2^	8
GO:0006862	nucleotide transport	2.41 × 10^−2^	8
GO:1902475	L-alpha-amino acid transmembrane transport	1.97 × 10^−2^	5
GO:1903825	organic acid transmembrane transport	4.85 × 10^−2^	11
GO:1905039	carboxylic acid transmembrane transport	4.85 × 10^−2^	11
GO:0015748	organophosphate ester transport	3.71 × 10^−2^	12
GO:0001889	liver development	4.33 × 10^−2^	6
GO:1901565	organonitrogen compound catabolic process	4.10 × 10^−2^	18
GO:0032774	RNA biosynthetic process	3.83 × 10^−2^	10
GO:2000113	negative regulation of cellular macromolecule biosynthetic process	4.64 × 10^−2^	19
GO:0045892	negative regulation of transcription, DNA-templated	3.94 × 10^−2^	15
GO:0051252	regulation of RNA metabolic process	3.47 × 10^−2^	44
CELLULAR COMPONENT			
GO:0005762	mitochondrial large ribosomal subunit	4.86 × 10^−2^	5
GO:0000315	organellar large ribosomal subunit	4.86 × 10^−2^	5
GO:0005840	ribosome	2.88 × 10^−2^	8
GO:0098687	chromosomal region	1.52 × 10^−2^	6
GO:1990904	ribonucleoprotein complex	2.32 × 10^−2^	14
GO:0000785	chromatin	6.42 × 10^−3^	8
MOLECULAR FUNCTION			
GO:0000295	adenine nucleotide transmembrane transporter activity	2.41 × 10^−2^	8
GO:0015216	purine nucleotide transmembrane transporter activity	2.41 × 10^−2^	8
GO:0015215	nucleotide transmembrane transporter activity	2.41 × 10^−2^	8
GO:0005346	purine ribonucleotide transmembrane transporter activity	2.65 × 10^−2^	7
GO:0015605	organophosphate ester transmembrane transporter activity	2.41 × 10^−2^	8
GO:0015932	nucleobase-containing compound transmembrane transporter activity	2.41 × 10^−2^	8
GO:1901505	carbohydrate derivative transmembrane transporter activity	2.65 × 10^−2^	7
GO:0008514	organic anion transmembrane transporter activity	4.34 × 10^−2^	13
GO:0008134	transcription factor binding	3.53 × 10^−2^	8
GO:0016772	transferase activity, transferring phosphorus-containing groups	1.40 × 10^−2^	9

## Data Availability

Data can be found in Figshare at 10.6084/m9.figshare.15157257.

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
