# Peer review of "Mitochondrial Disruption by Amyloid Beta 42 Identified by Proteomics and Pathway Mapping"

_cells, 2021, doi:10.3390/cells10092380_

Round 1
Reviewer 1 Report
In this article, the authors identified the mitochondrial disruption by amyloid-beta 42 through proteomics and pathway mapping. This study is comprehensive, which provided the importance of mitochondrial responses to Abeta toxicity. I recommend this paper be published at Cells after major revision.
- Figures 1 and 3 should be revised. For example, in Figure 1, Aβ was written as ‘A!’. Also, some labels in the chord graph from Figure 3C are hard to recognize. In addition, the quality of Figure 3B should be improved. Authors should carefully fix the typo and clarify the labels in the figures.
- In the abstract, authors should describe what NGF means.
- Supporting information was not provided. Authors should submit the supporting information for further review.
- The authors described the proteomic mechanisms of Abeta cellular calcium signaling in Figure 7. Can authors provide a specific explanation about the mitochondrial disruption caused by Abeta calcium-mediated?
Author Response
Reviewer 1
- Figures 1 and 3 should be revised. For example, in Figure 1, Aβ was written as ‘A!’. Also, some labels in the chord graph from Figure 3C are hard to recognize. In addition, the quality of Figure 3B should be improved. Authors should carefully fix the typo and clarify the labels in the figures.
Thank you for noticing. The issue here appears to arise during document conversion into PDF via the publisher’s submission site. We are in contact with the journal to correct this.
- In the abstract, authors should describe what NGF means.
We have added the full term nerve growth factor to the abstract (line 15).
- Supporting information was not provided. Authors should submit the supporting information for further review.
Supporting information regarding specific proteins and quantitative changes is included in Supplementary tables.
- The authors described the proteomic mechanisms of Abeta cellular calcium signaling in Figure 7. Can authors provide a specific explanation about the mitochondrial disruption caused by Abeta calcium-mediated?
At the reviewer’s suggestion we have elaborated on previous studies showing that A?42 is thought to mediate calcium entry through interactions with cell-surface receptors including NMDAR and ?7 nACh [12,13]. These receptors have been shown to impact mitochondrial activity and localization. This is stated on Pg 8 (lines 223 - 226) of the revised manuscript.
Reviewer 2 Report
The authors applied proteomics and informatic analysis to determine the cellular homeostatic changes in PC12 cells at chronic beta-amyloid treatment condition. They have identified changes in proteome with a higher alteration in mitochondrial proteins. The results are interesting and may be considered for publication once the quality of manuscript is improved by addressing following concerns.
- Dysfunction in autophagy-lysosomal pathway has been shown to be one of the earliest events in AD brains (Nixon et al, 2005; Gowrishankar et al, 2015; Sharoar et al, 2016 & 2021). I am wondering that proteome changes in this pathway did not pick in the authors’ results. ~1/3 proteome changes in beta-amyloid treatment cases were categorized as other organelles (Fig. 3B). Were there any autophagy-lysosomal proteins change? If there are considerable number of proteins in this category, authors may present it as separate category.
- It is not clear what the authors wanted to explain in 3.4 (page 8, line 195). The subtitle also not indicating a clear meaning. If the authors want to connect or emphasize that some know genetic disease conditions, such as 1p36 syndrome, are more susceptible to beta-amyloid effect, they may elaborate it in discussion. Infect, they discuss very shortly on their results in this section.
- There is no detail description on what type of beta-amyloid peptides (monomers/oligomers/fibrils) were used, how the peptides were prepared.
- Did the authors measure cell death for this treatment period?
Minor
- There are several typos and grammatical error throughout the manuscript that need to be corrected; e.g., page 1 line 29, page 2 line 69, page 3 line 109-110 119 124, page 4 line 135.
- Please change “statistically altered” to “significantly altered”.
Author Response
Reviewer 2
Dysfunction in autophagy-lysosomal pathway has been shown to be one of the earliest events in AD brains (Nixon et al, 2005; Gowrishankar et al, 2015; Sharoar et al, 2016 & 2021). I am wondering that proteome changes in this pathway did not pick in the authors’ results. ~1/3 proteome changes in beta-amyloid treatment cases were categorized as other organelles (Fig. 3B). Were there any autophagy-lysosomal proteins change? If there are considerable number of proteins in this category, authors may present it as separate category.
We thank the reviewer for the insight. Indeed, we also were looking for such processes however an analysis of the proteomic data favors the mitochondrial proteome. This may be due to several factors including cell type specific mechanisms of amyloid peptide activity. However, to address this point, we have revised the manuscript by adding the following sentence (Pg 9 line 250 - 255) in the discussion:
Interestingly, lysosome and autophagy specific proteins were not identified by our GO analysis as could be expected based on known early dysfunction in the autophagic-lysosomal pathway in AD [38]. This may be due to cell-specific responses to the amyloid peptide seen in PC12 cells. However, we identified 2 subunits of the biogenesis of lysosome-related organelle complex 1 (BLOC1S2 and BLOC1S6) that is involved in endosome recycling, another mechanism implicated in AD [39,40].
It is not clear what the authors wanted to explain in 3.4 (page 8, line 195). The subtitle also not indicating a clear meaning. If the authors want to connect or emphasize that some know genetic disease conditions, such as 1p36 syndrome, are more susceptible to beta-amyloid effect, they may elaborate it in discussion. Infect, they discuss very shortly on their results in this section.
Thank you for your comments. At the reviewer’s suggestion, we have elaborated on the findings. This can be seen on Pg 9 (lines 256-263) in the addition of the statement and the altered subtitle on Pg 8 (lines 197 - 198).
3.4. Genomic Enrichment suggests an involvement of A?42 proteome encoding genes in human disease
The GEA shows that 70% of the A?42 proteome encoding genes map to areas prone to copy number variations that result in conditions related to development including phenotypes such as microcephaly, facial dysmorphisms, and intellectual deficits. The results suggest that these genomic regions may be more vulnerable to the effects of A?42. It would be interesting to examine how other proteins encoded within these regions are involved with the normal developmental process of aging and how A?42 changes their expression and function in the context of AD and other amyloidogenic conditions that effect brain function including Parkinson’s and prion conditions.
There is no detail description on what type of beta-amyloid peptides (monomers/oligomers/fibrils) were used, how the peptides were prepared.
A statement has been added to describe our A?42 prep as previously described in the literature based on the source of the provided amyloid. Here the amyloid preparation was performed identically to methods shown to yield a stable oligomeric form. This is shown in the revised manuscript on Pgs. 2 (lines 67-69)
Soluble oligomeric A?42 peptides were prepared as described in Arora, K., et al [15]. These preparations have been shown to yield a stable oligomeric form of A?42
Minor
There are several typos and grammatical error throughout the manuscript that need to be corrected; e.g., page 1 line 29, page 2 line 69, page 3 line 109-110 119 124, page 4 line 135.
Thank you for pointing out these typos and grammatical errors which are now fixed.
Please change “statistically altered” to “significantly altered”.
This is changed.
Reviewer 3 Report
The authors treated differentiated PC12 cells with amyloid-beta as a cellular model of A-beta pathology in Alzheimer’s disease. By mass spectrometry they generated peptide fingerprints of such cells and of untreated controls. By comparison with a rat database they identified the corresponding protein s and significant differences in their abundance between A-beta treated and control cells defined an A-beta ptoteome. Extensive biometric analyses of this proteome revealed a dominance of proteins present in mitochondria (either exclusively or also as components of other cellular compartments) and suggested an impairment of mitochondrial intermediate and energy metabolism, membrane structures and partially calcium signaling. This is a sophisticated proteomic approach to estimate the mitochondrial involvement in A-beta pathology, which is debated since a long time and had also been elucidated by some proteomic approaches in patients, transgenic mice and A-beta treated primary neuron cultures earlier. I have only one major remark and a few minor suggestions.
Major remark
- The authors may describe more precisely their Aß42 preparation (e.g. source, in which solvent initially). This may also include the question, if some kind of pre-aggregation step was performed prior to adding Aß42 to the cell culture medium, to ensure eventually the presence of toxic oligomeric molecules. Because the pathology (and likely the changes of protein patterns) may depend on the aggregation state of the added Aß42 during the culture period. Formation of oligomers from commercial monomeric Aß may take some time. Was this somehow controlled ? Or was the intentiuon – in contrast- to apply monomeric Aß ? Important: Because I assume that the badly soluble Aß42 was first pre-solved in another medium, the authors should mention, if the ‘untreated’ PC12 cell controls were ‘solvent controls’, which received the same concentration of this solvent. Or was Aß42 directly diluted in cell culture medium ?
Minor issues
- Methods 2.1.: protease inhibitor tablet in which volume ?
- Methods 2.1 …at a flow rate was (of ?)…
- In several Figures some numbers/designations are too small to be well read in print
- Fig. 3B: not enough contrast between red letters and dark background in print
- Sub-headline 3.3: ‘changes’ instead of ‘ changed’ ?
Author Response
Reviewer 3
The authors treated differentiated PC12 cells with amyloid-beta as a cellular model of A-beta pathology in Alzheimer’s disease. By mass spectrometry they generated peptide fingerprints of such cells and of untreated controls. By comparison with a rat database they identified the corresponding protein s and significant differences in their abundance between A-beta treated and control cells defined an A-beta ptoteome. Extensive biometric analyses of this proteome revealed a dominance of proteins present in mitochondria (either exclusively or also as components of other cellular compartments) and suggested an impairment of mitochondrial intermediate and energy metabolism, membrane structures and partially calcium signaling. This is a sophisticated proteomic approach to estimate the mitochondrial involvement in A-beta pathology, which is debated since a long time and had also been elucidated by some proteomic approaches in patients, transgenic mice and A-beta treated primary neuron cultures earlier. I have only one major remark and a few minor suggestions.
Major remark
The authors may describe more precisely their Aß42 preparation (e.g. source, in which solvent initially). This may also include the question, if some kind of pre-aggregation step was performed prior to adding Aß42 to the cell culture medium, to ensure eventually the presence of toxic oligomeric molecules. Because the pathology (and likely the changes of protein patterns) may depend on the aggregation state of the added Aß42 during the culture period. Formation of oligomers from commercial monomeric Aß may take some time. Was this somehow controlled ? Or was the intentiuon – in contrast- to apply monomeric Aß ? Important: Because I assume that the badly soluble Aß42 was first pre-solved in another medium, the authors should mention, if the ‘untreated’ PC12 cell controls were ‘solvent controls’, which received the same concentration of this solvent. Or was Aß42 directly diluted in cell culture medium ?
A statement has been added to describe our A?42 prep as previously described in the literature based on the source of the provided amyloid. Here the amyloid preparation was performed identically to methods shown to yield a stable oligomeric form. This is shown in the revised manuscript on Pgs. 2 (lines 67-69)
Soluble oligomeric A?42 peptides were prepared as described in Arora, K., et al [15]. Daily media and treatment changes ensured consistent exposure to A?42 [16].
Minor issues
Methods 2.1.: protease inhibitor tablet in which volume ?
After three days cells were washed with cold 1X PBS then solubilized by the addition of a 0.1% TritonX-100 lysis buffer (TritonX-100, 1M Tris HCl, 1.5M NaCl, 0.25M EDTA, 10% glycerol, and 1 protease inhibitor tablet (Complete Mini, Roche) in 10ml of lysis buffer) at 4°C overnight.
Methods 2.1 …at a flow rate was (of ?)…
Peptides were separated using a reversed-phase PepMap RSLC 75 ?m i.d X 15 cm long with 2 ?m particle size C18 LC column (ThermoFisher Scientific), eluted with 0.1% formic acid and 80% acetonitrile at a flow rate of 300 nl/min. As indicated in the revised manuscript on Pg. 2 (Line 88)
In several Figures some numbers/designations are too small to be well read in print
Thank you for noticing. The issue here appears to arise during document conversion into PDF via the publisher’s submission site. We are in contact with the journal to correct this.
Fig. 3B: not enough contrast between red letters and dark background in print
Thank you for noticing. The issue here appears to arise during document conversion into PDF via the publisher’s submission site. We are in contact with the journal to correct this.
Sub-headline 3.3: ‘changes’ instead of ‘ changed’ ?
This is corrected.
Round 2
Reviewer 1 Report
The authors of this paper properly addressed the reviewer's question.
Reviewer 3 Report
The authors sufficiently answered my points.
Just a few hints for typos:
Line 549 (are) is known to mediate…
Line 604: various (type) types…
Line 871: at the H+ generating (complex) complexes…